# Prevalence and determinants of healthcare avoidance during the COVID-19 pandemic: A population-based cross-sectional study

Marije J. Splinter[1], Premysl Velek[1,2], M. Kamran Ikram[1,3], Brenda C. T. Kieboom[1,2], Robin P. Peeters[4], Patrick J. E. Bindels[2], M. Arfan Ikram[1], Frank J. Wolters[1,5], Maarten J. G. Leening[1,6], Evelien I. T. de Schepper[2], Silvan Licher[1]*

1 Department of Epidemiology, Erasmus MC—University Medical Center Rotterdam, Rotterdam, the Netherlands, 2 Department of General Practice, Erasmus MC—University Medical Center Rotterdam, Rotterdam, the Netherlands, 3 Department of Neurology, Erasmus MC—University Medical Center Rotterdam, Rotterdam, the Netherlands, 4 Department of Internal Medicine, Erasmus MC—University Medical Center Rotterdam, Rotterdam, the Netherlands, 5 Department of Radiology and Nuclear Medicine, Erasmus MC—University Medical Center Rotterdam, Rotterdam, the Netherlands, 6 Department of Cardiology, Erasmus MC—University Medical Center Rotterdam, Rotterdam, the Netherlands

* s.licher@erasmusmc.nl

**Data Availability Statement:** Data from The Rotterdam Study can be made available to interested researchers upon request. Requests can

## Abstract

### Background

During the Coronavirus Disease 2019 (COVID-19) pandemic, the number of consultations and diagnoses in primary care and referrals to specialist care declined substantially compared to prepandemic levels. Beyond deferral of elective non-COVID-19 care by healthcare providers, it is unclear to what extent healthcare avoidance by community-dwelling individuals contributed to this decline in routine healthcare utilisation. Moreover, it is uncertain which specific symptoms were left unheeded by patients and which determinants predispose to healthcare avoidance in the general population. In this cross-sectional study, we assessed prevalence of healthcare avoidance during the pandemic from a patient perspective, including symptoms that were left unheeded, as well as determinants of healthcare avoidance.

### Methods and findings

On April 20, 2020, a paper COVID-19 survey addressing healthcare utilisation, socioeconomic factors, mental and physical health, medication use, and COVID-19–specific symptoms was sent out to 8,732 participants from the population-based Rotterdam Study (response rate 73%). All questionnaires were returned before July 10, 2020. By hand, prevalence of healthcare avoidance was subsequently verified through free text analysis of medical records of general practitioners. Odds ratios (ORs) for avoidance were determined using logistic regression models, adjusted for age, sex, and history of chronic diseases. We found that 1,142 of 5,656 included participants (20.2%) reported having avoided healthcare. Of those, 414 participants (36.3%) reported symptoms that potentially warranted urgent evaluation, including limb weakness (13.6%), palpitations (10.8%), and chest pain (10.2%). Determinants related to avoidance were older age (adjusted OR 1.14 [95% confidence

be directed to the secretariat of the department of Epidemiology (secretariat.epi@erasmusmc.nl), or visit the following website for more information http://www.ergo-onderzoek.nl/wp/contact. We are unable to place data in a public repository due to legal and ethical restraints. Sharing of individual participant data was not included in the informed consent of the study, and there is potential risk of revealing participants' identities as it is not possible to completely anonymise the data.

**Funding:** SL and ES obtained a dedicated COVID-19 research grant from the Netherlands Organisation for Health Research and Development (ZonMw; 10430022010016). URL: https://www.zonmw.nl/en/. The Rotterdam Study is funded by the Erasmus University Medical Center and Erasmus University, Rotterdam; the Netherlands Organisation for Health Research and Development (ZonMw); the Research Institute for Diseases in the Elderly (RIDE); the Ministry of Education, Culture and Science; the Ministry of Heatlh, Welfare and Sport; the European Commission (DG XII); and the Municipality of Rotterdam. The funders had no role in study design, data collection and analysis, decision to publish, or preparation of the manuscript.

**Competing interests:** The authors have declared that no competing interests exist.

**Abbreviations:** CESD, Center for Epidemiological Studies Depression; CI, confidence interval; COVID-19, Coronavirus Disease 2019; GP, general practitioner; HADS, Hospital Anxiety and Depression Scale; ISCED, International Standard Classification of Education; OR, odds ratio; TIA, transient ischemic attack; UNESCO, United Nations Educational, Scientific, and Cultural Organization.

interval (CI) 1.08 to 1.21]), female sex (1.58 [1.38 to 1.82]), low educational level (primary education versus higher vocational/university 1.21 [1.01 to 1.46]), poor self-appreciated health (per level decrease 2.00 [1.80 to 2.22]), unemployment (versus employed 2.29 [1.54 to 3.39]), smoking (1.34 [1.08 to 1.65]), concern about contracting COVID-19 (per level increase 1.28 [1.19 to 1.38]) and symptoms of depression (per point increase 1.13 [1.11 to 1.14]) and anxiety (per point increase 1.16 [1.14 to 1.18]). Study limitations included uncertainty about (perceived) severity of the reported symptoms and potentially limited generalisability given the ethnically homogeneous study population.

## Conclusions

In this population-based cross-sectional study, 1 in 5 individuals avoided healthcare during lockdown in the COVID-19 pandemic, often for potentially urgent symptoms. Healthcare avoidance was strongly associated with female sex, fragile self-appreciated health, and high levels of depression and anxiety. These results emphasise the need for targeted public education urging these vulnerable patients to timely seek medical care for their symptoms to mitigate major health consequences.

## Author summary

### Why was this study done?

➤ During the Coronavirus Disease 2019 (COVID-19) pandemic, consultation rates in both primary and specialist care declined compared to prepandemic levels, which can partially be attributed to the postponement or cancellation of elective and nonurgent medical care.

➤ It is unclear to what extent these declines in consultation rates could be related to healthcare avoidance by patients in the general population.

➤ To evaluate the collateral health damage of the COVID-19 pandemic, it is important to not only assess the prevalence of healthcare avoidance, but also for what symptoms healthcare was avoided and which determinants are associated with this behaviour.

### What did the researchers do and find?

➤ We sent out a paper questionnaire to 8,732 participants of the population-based Rotterdam Study containing several COVID-19–related subjects, such as healthcare utilisation, work status, mental and physical health, and concerns about contracting COVID-19.

➤ About 6,241 participants (73%) returned the questionnaire, of whom 5,656 participants (90.6%) were included in our analyses. We found that 1,142 of them (20.2%) avoided healthcare during the COVID-19 pandemic, often for symptoms that might have needed urgent medical evaluation, such as limb weakness (13.6%), palpitations (10.8%), and chest pain (10.2%).

➤ Determinants that were most strongly associated with healthcare avoidance were female sex, poor self-appreciated health, and high levels of depression and anxiety.

**What do these findings mean?**

➤ The results of this population-based study suggest that healthcare avoidance contributed to the decline in consultation rates during the COVID-19 pandemic. Importantly, our findings suggest that this behaviour may be associated with certain vulnerable groups within the population.

➤ These findings should be interpreted in light of the limitations of this study, which include that the actual severity of the symptoms that were reported by participants is unknown, since they were not medically evaluated when they experienced these symptoms.

➤ The findings of this study can be used to develop policy interventions targeted to vulnerable individuals who may be more likely to exhibit healthcare avoiding behaviours.

## Introduction

In the first months of 2020, the number of confirmed COVID-19 cases in Europe began to rise, to which many European countries responded with restrictive measures aimed at limiting individual mobility in order to prevent overwhelming their healthcare systems [1–3]. On March 11, 2021, a year after COVID-19 was declared a pandemic by WHO, 40.5 million European citizens had been infected with the virus, and 904,100 had died [4]. During this year, the main focus was on facilitating acute medical care, while most scheduled and preventive care was cancelled or postponed [5,6]. Consequently, the number of consultations and diagnoses in primary care related to chronic diseases, such as cancer, cardiovascular diseases, and mental illnesses, as well as referrals to hospitals for these indications, declined in the first 6 months of 2020 compared to 2019 [7–11].

Thus far, the observed changes in healthcare utilisation are exclusively based on registry data of diagnoses instead of the actual symptoms experienced by individuals in the general population [7–9]. Detailed data on healthcare-seeking behaviour in primary care from a patient perspective provide complementary insights in the symptoms that were left unheeded and by whom. During the pandemic, especially during national lockdowns, individuals might refrain from seeing their general practitioner (GP) because they perceive their symptoms as too insignificant for burdening their physician, or not worth the risk of a COVID-19 infection [8,9]. Although many symptoms in primary care are self-limiting, urgent medical evaluation is essential for some in order to mitigate health damage. For example, symptoms that signal (transient) cardiovascular or cerebrovascular events could, if left untreated, lead to major health consequences. The collateral damage resulting from the pandemic is, therefore, not limited to patients who have been infected with COVID-19, but also affects vulnerable groups of individuals who experience difficulties or are afraid to access their primary care physician [12]. For this reason, the aim of this study is to expand our knowledge of healthcare avoidance in order to mitigate damage to population health in the aftermath of this or future pandemics, and of other disasters that could affect healthcare-seeking behaviour of citizens. Studies that so far focused on the patients' perspective in relation to healthcare avoidance identified several

groups at risk of avoiding healthcare. However, these studies have mainly been conducted in the United States, which do not have a primary care gatekeeper system as most European countries do [13–15]. This gatekeeper system provides a unique opportunity to meticulously assess changes in healthcare-seeking behaviour during the pandemic, since patients always have to contact their GP before they can be referred to a medical specialist [16].

In this cross-sectional study, embedded within an ongoing prospective cohort study, we determined prevalence of healthcare avoidance in the general population during the COVID-19 pandemic by combining self-reported healthcare-seeking behaviour with medical records of GPs. We also assessed the specific symptoms that were left unheeded, while specifically paying attention to the perception of the patient instead of the healthcare provider, and we sought to establish which potential determinants were associated with healthcare avoidance.

## Methods

### Study population

This cross-sectional study was embedded within the ongoing population-based Rotterdam Study, a prospective cohort study conceived to investigate the aetiology and natural history of chronic diseases in mid- and late-life [17]. The Rotterdam Study was initiated in 1990, when 7,983 residents of the district Ommoord in Rotterdam who were 55 years and older started their participation in the study [17]. Since then, the size of actively contributing, living participants remained largely stable, with 3 new study waves that have been initiated over time—while other participants passed away. In 2000, the cohort has been expanded with residents 55 years and older (RS-II, $N$ = 3,011). In 2006, 3,932 participants aged 45 years and older enrolled (R-III). In 2016, the most recent wave was initiated with 3,368 participants aged 40 and over contributing to the study (RS-IV). Since 1990, a total of 17,931 participants have taken part in the Rotterdam Study. All participants were extensively examined at study entry and subsequent follow-up every 3 to 6 years [17]. This study is reported as per the Strengthening the Reporting of Observational Studies in Epidemiology (STROBE) guideline (S1 Checklist). The prospective analysis plan of the current study has been included as a supporting information (S1 File).

### Data collection

For this cross-sectional study, we identified all participants that were still alive and actively taking part in the Rotterdam Study on April 8, 2020 ($N$ = 9,008). At that time, 8,732 participants (96.9%) were not hospitalised or living in nursing homes, thus included in the current study. In the Dutch healthcare system, residents of nursing homes are under direct and daily medical supervision of a geriatrician or nursing home physician, limiting potential healthcare avoidance. On April 20, 2020, we have invited the noninstitutionalised participants to fill out a paper COVID-19 questionnaire about the preceding period starting from the first confirmed COVID-19 infection in the Netherlands on February 27, 2020, which also indicated the start of the first wave of COVID-19. The questionnaire addressed healthcare utilisation, socioeconomic factors, mental and physical health, medication use, and COVID-19–specific symptoms. A detailed description of the methods including validation of the questionnaire has been reported elsewhere [18].

### Assessment of healthcare avoidance

Participants were asked to report whether they had experienced symptoms in the preceding weeks for which they otherwise would have contacted their GP or medical specialist but now did not do so because of COVID-19. They were also provided with a prespecified list of both potentially urgent symptoms (such as palpitations, chest pain, and limb weakness) and generic

symptoms (such as lower back pain) to enable them to indicate for which symptoms they had avoided healthcare. Since lower back pain is generally self-limiting, we have specifically used this symptom to contrast with other symptoms that might have required urgent medical evaluation. We subsequently checked the GP records by hand from December 2019 until December 2020 of all participants who reported having experienced symptoms for which they did not seek medical attention. The GP is generally the first to contact when an individual has symptoms. Visits to emergency departments or medical specialists are documented by the GP as well. In presence of self-reported symptoms in the questionnaire, healthcare avoidance would have, therefore, been reflected by the absence of physical, telephone, and administrative consultations in the medical records kept by the GP. These records contained narrative data, which means that the notes of the GP had been entered in a free text instead of structured format using prespecified diagnostic codes [19]. While the latter would have mainly included basic information such as laboratory results and patient demographics, narrative medical records are more accurate and detailed in scope, also including information on comorbidities, medication use, physical exams, the GP's impression, and treatment plan [19]. Analysis of these records resulted in a detailed overview of the healthcare-seeking behaviour of healthcare avoiding participants. We defined 3 levels of certainty of healthcare avoidance: "definite," "probable," and "possible." Participants who did not contact their GP for the symptoms they had mentioned on the questionnaire were definite healthcare avoiders. In case they had reached out to their GP more than 2 weeks after they had filled out the questionnaire, indicating a delay in healthcare-seeking behaviour, they were considered a probable healthcare avoider. The remaining participants, who had had contact with their GP despite reporting themselves as healthcare avoiders, were labelled as possible healthcare avoiders. The GP records also gave us the opportunity to compare consultation rates between 3 control months prior to the first wave of COVID-19, with consultation rates during the months of the lockdown itself.

## Determinants related to healthcare avoidance

Based on literature, we have prespecified the following determinants of healthcare avoidance for inclusion in the questionnaire: age, sex, self-appreciated health (excellent; very good; good; fair; poor), occupational status (working; on sick leave; unemployed; retired; other), alcohol consumption and smoking status (self-reported use during the 14 days prior to filling out the questionnaire), concern about contracting COVID-19 (never; rarely; sometimes; often; almost continuously), depression (weighted score on 10 out of 20 questions from the Center for Epidemiological Studies Depression (CESD) scale, with a maximum score of 29), and anxiety (weighted score on 7 out of 14 questions from the Hospital Anxiety and Depression Scale (HADS), with a maximum score of 20) [9,10,13,20–24]. We have also asked participants about their medical history, including a history of chronic diseases (such as cancer; heart disease; stroke; chronic lung disease; neurodegenerative disease; diabetes; mental illness). The educational level of participants (primary education; low/intermediate general or lower vocational; intermediate vocational or higher general; higher vocational or university) was retrieved from earlier measurements in 2015 (cohorts I, II, and III) and 2020 (cohort IV), according to the International Standard Classification of Education (ISCED) by the United Nations Educational, Scientific, and Cultural Organization (UNESCO) [25].

## Statistical analyses

Characteristics of the study population that were measured on a continuous scale were represented by the mean and standard deviation, whereas categorical variables were presented as

the total number of observations with corresponding percentages. Age was subdivided into 3 categories to calculate age-specific prevalence of healthcare avoidance. Missing values in determinants (all less than 1.3% missing) were imputed using the fully conditional specification method with a maximum number of 10 iterations. We did not find evidence for issues of multicollinearity.

We have employed binary logistic regression analyses to assess the association between determinants and healthcare avoidance. These analyses were conducted in 3 different steps. First, we have investigated the association between a particular determinant, 2 confounders (age and sex), and healthcare avoidance (model 1). Then, we have repeated these analyses while adding another confounder to the model, which was a history of self-reported chronic diseases (model 2). Finally, we have conducted multivariable logistic analyses adjusting for all considered determinants in this study (model 3). Results were presented as odds ratios (ORs) with corresponding 95% confidence intervals (CIs). To evaluate the robustness of our findings, we have conducted 4 sensitivity analyses. First, we have stratified main analyses between participants who reported symptoms that might have warranted urgent medical assessment (chest pain, palpitations, limb weakness, difficulty speaking or facial drooping, and self-perceived cancer-related symptoms) and the remaining symptoms of a more generic nature that were listed in the questionnaire (lower back pain, sudden onset dizziness, memory complaints, fluid retention (oedema), elevated blood pressure, attempts to stop or reduce smoking, nausea or vomiting, sudden (temporary) vision loss, and dysregulation of diabetes). Second, we explored healthcare avoidance among participants that would most likely differ in their healthcare utilisation behaviour due to comorbidities by stratifying individuals with or without a history of any chronic disease. Third, we have compared the main analyses of definite and probable to possible healthcare avoiders to verify whether effect estimates would differ between these different levels of healthcare avoidance. Finally, as a result of the peer review process, we have additionally stratified the analyses between the self-reported chronic diseases included in this study to examine whether the strength of the associations would differ depending on the type of disease. Data were handled and analysed with the Statistical Package for the Social Sciences software (SPSS), version 25.0. Level of statistical significance (alpha) was set at 0.05.

## Details of ethical approval

The Rotterdam Study has been approved by the Medical Ethics Committee of the Erasmus MC (registration number MEC 02.1015) and by the Dutch Ministry of Health, Welfare, and Sport (Population Screening Act WBO, license number 1071272-159521-PG). The Rotterdam Study has been entered into the Netherlands National Trial Register (NTR; www.trialregister. nl) and into the WHO International Clinical Trials Registry Platform (ICTRP; www.who.int/ ictrp/network/primary/en/) under shared catalogue number NTR6831. All participants provided written informed consent to participate in the study and to have their information obtained from treating physicians.

## Results

### Characteristics

The response rate of the questionnaire was 73% ($N = 6,241$). All questionnaires were returned before July 10, 2020. We excluded 9.3% ($N = 585$) that did not have complete data on the questions concerning healthcare utilisation. These participants were more often women, were of a lower educational level, and less often had a chronic disease (Table A in S1 Tables). Nonresponders to the questionnaire had a comparable mean age and ethnic background to responders, were slightly more often female and of a lower educational level (Table B in S1 Tables).

**Table 1. Characteristics of the study population (N = 5,656). Values are numbers (percentages) unless stated otherwise.**

| | | Population total (N = 5,656) | Nonavoiders (N = 4,514) | Avoiders (N = 1,142) |
|---|---|---|---|---|
| Age, years (mean, SD) | | 69.4 (11.5) | 68.8 (11.3) | 71.7 (11.9) |
| Age categories | <65 years | 1,880 (33.2) | 1,562 (34.6) | 318 (27.8) |
| | 65–79 years | 2,589 (45.8) | 2,106 (46.7) | 483 (42.3) |
| | ≥80 years | 1,187 (20.9) | 846 (18.7) | 341 (29.9) |
| Women | | 3,266 (57.7) | 2,508 (55.6) | 758 (66.4) |
| History of chronic diseases | Any | 3,661 (64.7) | 2,772 (61.4) | 889 (77.8) |
| | Cancer | 812 (14.4) | 601 (13.3) | 211 (18.5) |
| | Heart disease | 1,640 (29.0) | 1,224 (27.1) | 416 (36.4) |
| | Stroke | 418 (7.4) | 292 (6.5) | 126 (11.0) |
| | Chronic lung disease | 795 (14.1) | 569 (12.6) | 226 (19.8) |
| | Neurodegenerative disease | 97 (1.7) | 68 (1.5) | 29 (2.5) |
| | Diabetes | 547 (9.7) | 391 (8.7) | 156 (13.7) |
| | Mental illness | 257 (4.5) | 154 (3.4) | 103 (9.0) |
| Educational level | Primary education | 343 (6.1) | 241 (5.3) | 102 (8.9) |
| | Low/intermediate general or lower vocational | 1,875 (33.2) | 1,454 (32.2) | 421 (36.9) |
| | Intermediate vocational or higher general | 1,807 (31.9) | 1,452 (32.2) | 355 (31.1) |
| | Higher vocational or university | 1,579 (27.9) | 1,330 (29.5) | 249 (21.8) |
| Self-appreciated health | Poor | 62 (1.1) | 24 (0.5) | 38 (3.4) |
| | Fair | 731 (12.9) | 433 (9.6) | 298 (26.7) |
| | Good | 3,197 (56.5) | 2,577 (57.1) | 620 (55.6) |
| | Very good | 1,153 (20.4) | 1,029 (22.8) | 124 (11.1) |
| | Excellent | 416 (7.4) | 380 (8.4) | 36 (3.2) |
| Occupational status | Working (full time, part-time, self-employed) | 1,578 (27.9) | 1,375 (30.5) | 203 (17.8) |
| | On sick leave | 61 (1.1) | 45 (1.0) | 16 (1.4) |
| | Unemployed | 161 (2.8) | 117 (2.6) | 44 (3.9) |
| | Retired | 3,407 (60.2) | 2,684 (59.5) | 723 (63.3) |
| | Other | 289 (5.1) | 197 (4.4) | 92 (8.1) |
| Alcohol consumption; yes | | 3,092 (54.7) | 2,545 (56.4) | 547 (47.9) |
| Current smoking; yes | | 591 (10.4) | 456 (10.1) | 135 (11.8) |
| Concern contracting COVID-19 | Never | 893 (15.8) | 742 (16.4) | 151 (13.2) |
| | Rarely | 1,766 (31.2) | 1,486 (32.9) | 280 (24.5) |
| | Sometimes | 2,417 (42.7) | 1,910 (42.3) | 507 (44.4) |
| | Often | 446 (7.9) | 296 (6.6) | 150 (13.1) |
| | Almost continuously | 76 (1.3) | 44 (1.0) | 32 (2.8) |
| Symptoms of depression (weighted score ≥ 10) | | 911 (16.1) | 554 (12.3) | 357 (31.3) |
| Symptoms of anxiety (weighted score ≥ 7) | | 889 (15.7) | 549 (12.2) | 340 (29.8) |

CI, confidence interval; COVID-19, Coronavirus Disease 2019; N, number of participants; SD, standard deviation.

The final population size consisted of 5,656 individuals. In Table 1, it is shown that 1,142 participants (20.2%) have avoided healthcare despite experiencing symptoms. Most of those participants considered to be healthcare avoiders (42.3%) were in the age category 65 to 79 years,

**Table 2. Symptoms for which healthcare was avoided (*N* = 1,142)*.**

| | | *N* (%) |
|---|---|---|
| Palpitations | All | 123 (10.8) |
| | Among those with a history of CVD | 75 (60.9) |
| Chest pain | All | 116 (10.2) |
| | Among those with a history of CVD | 57 (49.0) |
| Limb weakness | | 155 (13.6) |
| Self-perceived cancer-related symptoms | | 131 (11.5) |
| Difficulty speaking or facial drooping | | 23 (2.0) |
| Sudden (temporary) vision loss | | 51 (4.5) |
| Elevated blood pressure | | 101 (8.8) |
| Sudden onset dizziness | | 197 (17.3) |
| Dysregulation of diabetes | | 49 (4.3) |
| Nausea or vomiting | | 54 (4.7) |
| Fluid retention (oedema) | | 139 (12.2) |
| Memory complaints | | 177 (15.5) |
| Attempts to stop or reduce smoking | | 65 (5.7) |
| Lower back pain** | | 369 (32.3) |

CVD, cardiovascular disease; *N*, number of participants.

*1,142 = total number of healthcare avoiders.

**10.4% of the participants (*N* = 119) reported lower back pain as their only symptom.

36.3% of the participants (*N* = 414) reported at least one symptom, which should have received direct medical attention (palpitations, chest pain, limb weakness, self-perceived cancer-related symptoms, difficulty speaking).

54.1% of the participants (*N* = 618) reported one symptom; 44.4% (*N* = 508) reported 2 or more symptoms; 1.5% (*N* = 16) did not specify symptoms.

compared to 27.8% below age 65 and 29.9% from the age of 80 onwards. Healthcare avoiders were more likely to be women (66.4%) and to have a history of any chronic disease (77.8%).

## Symptoms and healthcare avoidance

Out of 1,142 healthcare avoiders, 16 did not specify their symptoms in the questionnaire, which means that the largest part of the study population (99.7%) included symptomatic patients who did not seek medical care. More than a third (36.3%) of all healthcare avoiding participants reported symptoms, which might have required urgent medical attention, with limb weakness (13.6%), self-perceived cancer-related symptoms (11.5%), palpitations (10.8%), and chest pain (10.2%) being most prevalent (Table 2). Respectively, 60.9% and 49.0% of participants who reported palpitations and chest pain had a history of cardiovascular disease. The high prevalence of lower back pain was mainly driven by a combination of additional symptoms, given that only 119 participants reported lower back pain as the only symptom for which they avoided healthcare.

## Analysis of GP records

We were able to review the records from 889 out of 1,142 (77.8%) participants who reported that they had avoided healthcare. The remaining 253 (22.2%) medical records could not be retrieved because participants had either not given consent to review their records, or they moved out of the Ommoord district after enrolment in the Rotterdam study, and, therefore, their records were not digitally accessible. Fig 1 shows that most participants met criteria to be

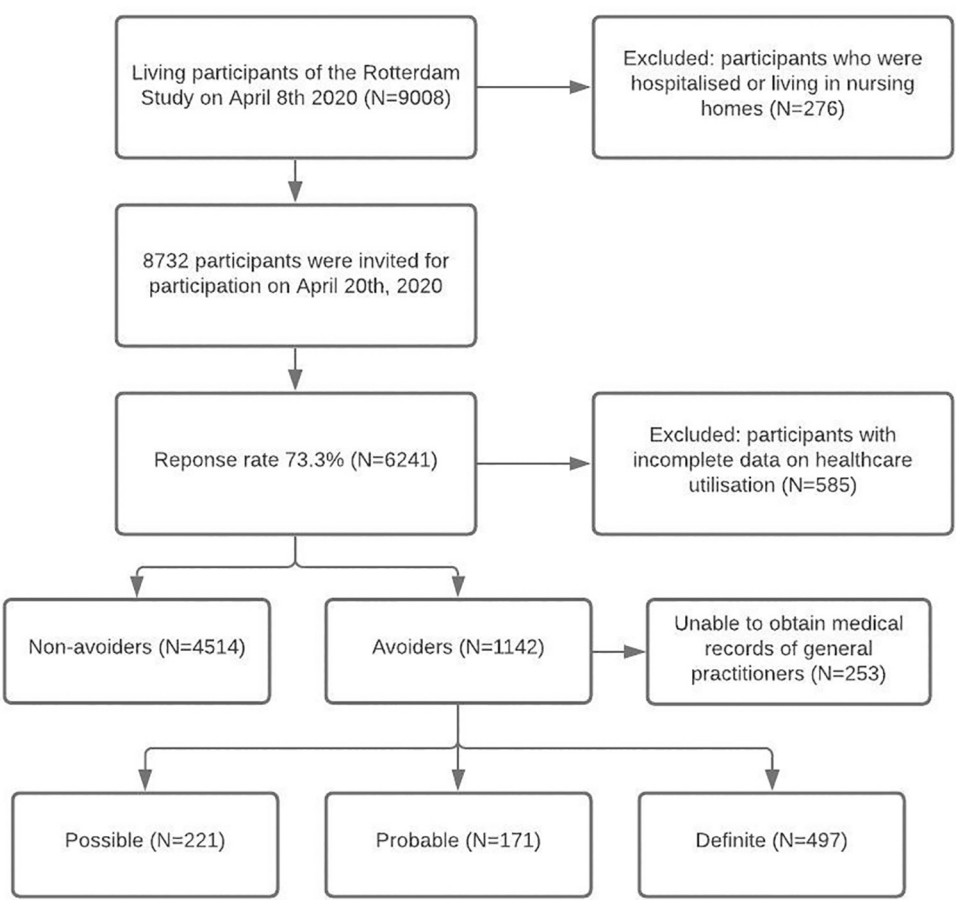

**Fig 1. Flow chart of the study population.** *N*, number of participants.

a definite healthcare avoider (*N* = 497). A minority was considered a probable healthcare avoider (*N* = 171), while possible healthcare avoiders (*N* = 221) were more prevalent. From the GP records, we also observed that the number of physical consultations plummeted during the first wave of COVID-19 compared to the 3 months prior to the pandemic (44.5% versus 66.2% of all GP consultations).

## Determinants related to healthcare avoidance

ORs for healthcare avoidance were higher among older participants (OR per 10 years increase 1.22 [1.15 to 1.29]; Table 3) and women (1.59 [1.38 to –1.82]). In age- and sex-adjusted models, low compared to high educational attainment was associated with healthcare avoidance (primary education versus higher vocational/university level 1.85 [1.39 to 2.46]). Moreover, the odds for avoidance were higher for participants with poor self-appreciated health (per point decrease 2.13 [1.93 to 2.35]), and for those who were unemployed compared to those who were employed (2.37 [1.60 to 3.51]). Retirement was not related to healthcare avoidance, after accounting for age and sex. Alcohol consumption was associated with a lower OR for avoidance (0.78 [0.68 to 0.89]), and an inverse relationship was found for current smokers (1.35 [1.09 to 1.66]). Increasing concern about contracting COVID-19 was related to higher ORs of healthcare avoidance (per level increase 1.33 [1.24 to 1.43]). Higher ORs were also seen for those who reported symptoms of depression (per point increase 1.13 [1.12 to 1.15]) or anxiety

**Table 3. Determinants of healthcare avoidance (N = 5,656).**

| | | Model 1 | Model 2 | Model 3 |
|---|---|---|---|---|
| | | OR (95% CI) | OR (95% CI) | OR (95% CI) |
| Age, per 10 years increase[a] | | 1.22 (1.15–1.29)** | 1.14 (1.08–1.21)** | 1.01 (0.93–1.10) |
| Women[b] | | 1.59 (1.38–1.82)** | 1.58 (1.38–1.82)** | 1.24 (1.06–1.46** |
| Educational level vs. higher vocational or university | Primary education | 1.85 (1.39–2.46)** | 1.21 (1.01–1.46)* | 1.12 (0.79–1.58) |
| | Low/intermediate general or lower vocational | 1.26 (1.04–1.51)* | 1.22 (1.01–1.46)* | 0.99 (0.80–1.22) |
| | Intermediate vocational or higher general | 1.24 (1.03–1.48)* | 1.21 (1.01–1.46)* | 1.08 (0.89–1.31) |
| Self-appreciated health, per level decrease | | 2.13 (1.93–2.35)** | 2.00 (1.80–2.22)** | 1.58 (1.41–1.76)** |
| Occupational status vs. employed | Retired | 1.26 (0.99–1.61) | 1.18 (0.92–1.51) | 1.34 (1.02–1.76)* |
| | Unemployed | 2.37 (1.60–3.51)** | 2.29 (1.54–3.39)** | 1.38 (0.88–2.17) |
| Alcohol consumption | | 0.78 (0.68–0.89)** | 0.81 (0.71–0.92)** | 0.90 (0.78–1.05) |
| Smoking | | 1.35 (1.09–1.66)** | 1.34 (1.08–1.65)** | 1.20 (0.95–1.52) |
| Concern contracting COVID-19, per level increase | | 1.33 (1.24–1.43)** | 1.28 (1.19–1.38)** | 1.00 (0.91–1.10) |
| Level of depression, per score increase | | 1.13 (1.12–1.15)** | 1.13 (1.11–1.14)** | 1.08 (1.05–1.11)** |
| Level of anxiety, per score increase | | 1.17 (1.14–1.19)** | 1.16 (1.14–1.18)** | 1.04 (1.01–1.08)* |

CI, confidence interval; COVID-19, Coronavirus Disease 2019; N, number of participants; OR, odds ratio.

[a]adjusted for sex.

[b]adjusted for age.

*$p < 0.05$

**$p < 0.01$.

Model 1: binary logistic regression analyses adjusted for age and sex.

Model 2: the same as model 1, additionally adjusted for a history of self-reported chronic diseases.

Model 3: multivariable logistic analyses adjusted for all determinants presented in Table 3.

(per point increase 1.17 [1.14 to 1.19]). Additional adjustment for a history of any chronic disease only slightly weakened results. Effect estimates further attenuated in models that were adjusted for all other considered potential determinants, yet largely remained direction consistent.

## Sensitivity analyses

Except for smoking, determinants were most strongly related to healthcare avoidance among participants who reported potentially alarming symptoms compared to those who only reported generic symptoms (Table C in S1 Tables). All determinants were more strongly associated with healthcare avoidance among participants with a history of any chronic disease compared to those without chronic diseases (Table D in S1 Tables). Determinants were stronger related to healthcare avoidance among definite and probable avoiders than possible avoiders, except for concern about contracting COVID-19, which showed comparable ORs (Table E in S1 Tables). Finally, self-appreciated health, concern about contracting COVID-19, and the level of depression and anxiety appeared to be strongly associated with healthcare avoidance among all chronic diseases except neurodegenerative diseases (Table F in S1 Tables).

## Discussion

In this cross-sectional study, we found that 1 out of every 5 individuals reported having avoided healthcare during lockdown of the COVID-19 pandemic. Among those, more than a third experienced symptoms that might have warranted urgent medical evaluation, with limb

weakness, self-perceived cancer-related symptoms, palpitations, and chest pain being the most prevalent. In multivariable analyses, we have shown that female sex, low self-appreciated health, and high levels of anxiety and depression were associated with healthcare avoidance during the COVID-19 pandemic.

## Comparison with previous studies

Previous studies revealed a global trend of declining diagnoses recorded by GPs and a substantial reduction of hospital admissions for acute coronary syndromes, strokes, and transient ischemic attacks (TIAs) during the first wave of COVID-19 [6,8–11,13,26–29]. Our analyses showed that a substantial part of the general population avoided healthcare for symptoms potentially indicating such cardiovascular or cerebrovascular diseases, which can have serious health damaging consequences on both short- and long-term. For instance, the 30-day risk of stroke or other vascular events after a TIA ranges from 3.2% to 17.7%, and the 5-year risk is approximately 6.4% [30,31]. Therefore, healthcare avoidance among participants who reported chest pain, palpitations, or limb weakness is particularly concerning. This does not implicate that healthcare avoidance among participants who experienced atypical symptoms, such as sudden dizziness, vision loss, nausea, or vomiting, is less severe, since these symptoms could signal underlying chronic conditions as well [9].

Several studies have theorised about explanatory mechanisms behind healthcare avoidance during the COVID-19 pandemic. For example, the so-called COVID Stress Syndrome proposes confidence in one's physical health to be able to overcome a COVID-19 infection as a determinant of healthcare avoidance [21,22]. Individuals with poor perceived health would prefer to avoid physical contact because of their concerns for a severe course of a COVID-19 infection [21]. Our finding that poor self-appreciated health was strongly associated with healthcare avoidance might support this hypothesis. Contrary to what would be expected based on literature [6,8–10], our study showed that self-appreciated health was more strongly associated with healthcare avoidance than concern about contracting COVID-19. This might be explained by the fact that we sent out the questionnaire during the first months of the pandemic, when the potential severity of COVID-19 was not as widely known as it is now.

## Strengths and limitations

One of the major strengths of this study is the direct, patient-centred approach. We managed to retrieve self-reported data on healthcare avoidance instead of concentrating on medical records of patients who have already been hospitalised or who visited an outpatient clinic. Moreover, we were able to complement our findings with GP records of a substantial part of the healthcare avoiding participants. Several limitations of this study must also be acknowledged. First, we were unable to assess the actual severity of the symptoms that participants reported in the questionnaire, because they were not medically evaluated at the time. Second, it is unknown how severe participants themselves perceived their symptoms to be, which could have affected their decision whether or not to seek medical attention [14]. Third, to minimise potential selection bias, we have first shown that responders and nonresponders had comparable characteristics, yet the ethnic homogeneity and higher educational attainment of the study population will limit generalisability of our study results to populations with more ethnic diversity or lower educational level. Fourth, participants could have interpreted the question on healthcare avoidance differently, as we asked them to base their responses on the weeks prior to filling out the questionnaire. Nevertheless, more than 90% of our study population (N = 5,151) returned the questionnaire before May 11. From this day onwards, several counter-measures that had been implemented by the Dutch government in March 2020 were lifted,

which means that most participants filled out the questionnaire while all of these countermeasures were still present.

## Implications

Collectively, findings of our study suggest that healthcare avoidance during COVID-19 may be prevalent among those who are in greater need of it in the population, such as older individuals, those with low perceived health, and those who report symptoms of poor mental health. These findings call for population-wide campaigns urging individuals who are most prone to avoid healthcare to timely reach out to their GP or medical specialist to report both alarming and seemingly insignificant symptoms. Furthermore, physicians should be made aware of which of their patients are most at risk of avoiding healthcare so that they can take a proactive role in approaching these patients, especially now that vaccination strategies are successfully being implemented and regular healthcare is gradually restarting [32].

## Conclusions

During lockdown in the COVID-19 pandemic, 1 out of 5 individuals in the general population avoided healthcare despite having symptoms. Female sex, fragile self-appreciated health, and high levels of depression and anxiety are particularly associated with healthcare avoidance, often for symptoms that might have required urgent medical assessment. Ongoing longitudinal tracking of the incidence of diseases in this study population will allow quantification of the exact magnitude of collateral health damage due to healthcare avoidance during the COVID-19 pandemic. Future studies should examine healthcare-seeking behaviour among ethnically diverse populations, which remain understudied.

## Supporting information

**S1 Checklist. STROBE checklist for cross-sectional studies.**
(DOCX)

**S1 File. Prospective Analysis Plan.**
(DOCX)

**S1 Tables. Supporting tables.** Table A. Characteristics of excluded participants. Table B. Characteristics of responders versus nonresponders. Table C. Determinants of healthcare avoidance stratified by potentially alarming and generic symptoms. Table D. Determinants of healthcare avoidance among participants with or without a history of any chronic disease. Table E. Determinants of healthcare avoidance stratified by different levels of healthcare avoidance. Table F. Determinants of healthcare avoidance stratified by chronic disease.
(DOCX)

## Acknowledgments

We acknowledge the dedication, commitment, and contribution of inhabitants, general practitioners, and pharmacists of the Ommoord district who took part in the Rotterdam Study. We acknowledge Frank van Rooij as data manager and thank Jolande Verkroost-van Heemst and Natalie Terzikhan for their invaluable contribution to the collection of the data. We are grateful to the valuable contribution of patient representatives from the Dutch Cancer Society and Harteraad to the design of the questionnaire.

## Author Contributions

**Conceptualization:** Marije J. Splinter, Premysl Velek, M. Kamran Ikram, Brenda C. T. Kieboom, Robin P. Peeters, Patrick J. E. Bindels, M. Arfan Ikram, Frank J. Wolters, Maarten J. G. Leening, Evelien I. T. de Schepper, Silvan Licher.

**Data curation:** Brenda C. T. Kieboom.

**Formal analysis:** Marije J. Splinter.

**Funding acquisition:** Evelien I. T. de Schepper, Silvan Licher.

**Investigation:** Marije J. Splinter.

**Methodology:** Marije J. Splinter, M. Kamran Ikram, Evelien I. T. de Schepper, Silvan Licher.

**Project administration:** M. Kamran Ikram, Evelien I. T. de Schepper, Silvan Licher.

**Resources:** Marije J. Splinter, M. Kamran Ikram, Frank J. Wolters, Maarten J. G. Leening, Evelien I. T. de Schepper, Silvan Licher.

**Supervision:** M. Kamran Ikram, Evelien I. T. de Schepper, Silvan Licher.

**Validation:** M. Kamran Ikram.

**Visualization:** Marije J. Splinter.

**Writing – original draft:** Marije J. Splinter.

**Writing – review & editing:** Marije J. Splinter, Premysl Velek, M. Kamran Ikram, Brenda C. T. Kieboom, Robin P. Peeters, Patrick J. E. Bindels, M. Arfan Ikram, Frank J. Wolters, Maarten J. G. Leening, Evelien I. T. de Schepper, Silvan Licher.

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
