## [Editor Report · Decision Letter 0]

12 Jul 2021

Dear Dr Splinter, 

Thank you for submitting your manuscript entitled "Prevalence and determinants of healthcare avoidance during the COVID-19 pandemic: a population-based study" for consideration by PLOS Medicine.

Your manuscript has now been evaluated by the PLOS Medicine editorial staff and I am writing to let you know that we would like to send your submission out for external peer review.

Please re-submit your manuscript within two working days, i.e. by Jul 14 2021 11:59PM.

Kind regards,

Callam Davidson

Associate Editor

PLOS Medicine

---

## [Decision Letter · Decision Letter 1]

14 Sep 2021

Dear Dr. Splinter,

Thank you very much for submitting your manuscript "Prevalence and determinants of healthcare avoidance during the COVID-19 pandemic: a population-based study" (PMEDICINE-D-21-02940R1) for consideration at PLOS Medicine. 

Your paper was evaluated by an associate editor and discussed among all the editors here. It was also discussed with an academic editor with relevant expertise, and sent to independent reviewers, including a statistical reviewer. The reviews are appended at the bottom of this email and any accompanying reviewer attachments can be seen via the link below:

[LINK]

In light of these reviews, we will not be able to accept the manuscript for publication in the journal in its current form, but we would like to consider a revised version that addresses the reviewers' and editors' comments. You will understand that we cannot make any decision about publication until we have seen the revised manuscript and your response, and we plan to seek re-review by one or more of the reviewers. 

We hope to receive your revised manuscript by Oct 05 2021 11:59PM. Please email us (plosmedicine@plos.org) if you have any questions or concerns.

We look forward to receiving your revised manuscript. 

Sincerely,

Callam Davidson, 

Associate Editor

PLOS Medicine

plosmedicine.org

Your study is observational and therefore causality cannot be inferred. Please remove language that implies causality, such as ‘Vulnerable patients avoided healthcare most’. Refer to associations instead (e.g. fragile self-appreciated health and poor socioeconomic status were associated with healthcare avoidance’). Please check throughout the manuscript. 

Please revise your title to include the study design. ‘Prevalence and determinants of healthcare avoidance during the COVID-19 pandemic: a population-based cross-sectional study’, or similar, would be appropriate.

The URL provided in your Data Availability Statement (in your response to the submission form) does not appear to be functioning, please check and correct as necessary. 

Please include line numbering in the margin of your manuscript to facilitate the review process.

Please structure your abstract using the PLOS Medicine headings (Background, Methods and Findings, Conclusions).

Please combine the Methods and Findings sections into one section, “Methods and findings”.

Please ensure that all numbers presented in the abstract are present and identical to numbers presented in the main manuscript text.

Please include the study design (cross-sectional), dates that the survey was sent out/returned, and expand on the content of the ‘COVID-19 survey’ in the abstract.

In the last sentence of the Abstract Methods and Findings section, please describe the main limitation(s) of the study's methodology.

Citations should be numerical, in square brackets, and preceding punctuation (e.g. [1]). Please update throughout.

In the final paragraph of the introduction, you refer to the study as a prospective cohort study. While the study was embedded in a prospective cohort study, the design of this study is cross-sectional. Please update as appropriate. 

Please review your methods section and ensure that methods are described in the past tense throughout. 

Some of the content presented in the methods section (e.g. response rates) may be better located in the results section.

In the results section titled ‘Determinants related to healthcare avoidance’, please remove the term ‘statistically insignificant’ and update to ‘the effect estimates of alcohol consumption and smoking were no longer statistically significant’.

It appears that the analysis you have performed should be described as multivariable rather than multivariate (see http://www.ncbi.nlm.nih.gov/pmc/articles/PMC3518362/ for definitions); please edit accordingly.

Please rearrange the Discussion such that the ‘Strengths and limitations’ section follows the ‘Comparison with previous studies’ section (and update numbering of citations as appropriate).

Please remove the ‘Competing interest statement’, ‘Transparency declaration’, ‘Details of funding’, ‘Role of the funding source’, ‘Statement of independence of researchers from funders’, ‘Contributors’, and ‘Data sharing statement’ from the end of the main text. In the event of publication, all of this information will be published as metadata based on your responses to the questions in the submission form. 

Please relocate the ‘Details of ethical approval’ section from the end of the main text to the methods section.

Please remove italicised formatting from your references and only use et al after listing the first six authors of a paper. See our website for other reference guidelines https://journals.plos.org/plosmedicine/s/submission-guidelines#loc-references

Thank you for providing your STROBE checklist. Please replace the page numbers with paragraph numbers per section (e.g. "Methods, paragraph 1"), since the page numbers of the final published paper may be different from the page numbers in the current manuscript.

Did your study have a prospective protocol or analysis plan? Please state this (either way) early in the Methods section.

Comments from the reviewers:

Reviewer #1: 

Authors should be attent to some things to be explaine more clearly related to :

* Study design; is not prospective cohort study but it is cross sectional survey, that was apart of the ongoing population-based Rotterdam Study, a prospective cohort study.

* using primary and secondary data, it is ok, but which one is going to be used? or is combine data going to be used as an eligibility? criteria of the study.?

* It needs clearly eligibility of the study for sampling such as age, not hospitalised or living in nursing homes, etc, it's has been written,more accurate in methods as a sample criteria of the study.

* Was questionare valideted? should be explained.

* Respons rate is better to write in the result of the study not in the methods

* Result, discussion, and conclusion was excellent…. added suggestions in conclusion is better. so that immediate action is taken regarding the benefits 

Reviewer #2: I confine my remarks to statistical aspects of this paper. The general approach is fine, but I do have a couple issues to resolve before I can recommend publication.

On page 6 the authors say they categorized age. This is a mistake. Categorizing continuous variables increases type 1 and type 2 error and introduces a kind of "magical thinking" that something special happens at the cut points. Leave age continuous and use splines to investigate nonlinearity. I wrote a paper on this: https://medium.com/@peterflom/what-happens-when-we-categorize-an-independent-variable-in-regression-77d4c5862b6c

Also on page 6, please provide some details on how the multiple imputation was done.

Table 1 Please give the mean and sd for age. (Or median and IQR)

Peter Flom

Reviewer #3: The study is interesting. The concern of health care avoidance was striking at the beginning of the pandemic as evident by the occupancy of the hospital beds that reduced to less than 1/3rd. The authors did a commendable job with the study. The response rate was 73% which is very good for this kind of study, and helped with the robustness of the data. 

I have few suggestions/queries:

MAJOR:

1. Page 5, Statistical analysis: I find the Logistic regression models bit confusing. It is stated that model-1 was adjusted for age and sex, but in the table I see there are more predictor variables (controls) than age and sex and same for model-2. Also for model-3, all other considered determinants need to be explained. How were those "considered determinants" considered appropriate for the model-3. It is better for the reliability of the data of authors explained how they chose the predictor variables for LR models. Also please add the Area under ROC for the models with confidence intervals. 

MINOR:

2. Aim is missing. 

3. In abstract: worry about contracting COVID-19� Better to change 'worry' with concern. 

4. Page 4: On April 8th 2020, 9008 out of a total of 18924 participants (47.6%) were still alive. � In 2016 it was 18924, and within 4 years only 47% from the database were alive? Any explanation for this or this is an error in reporting?

5. Page 4 and 5: "Assessment of healthcare avoidance" and "Determinants related to healthcare avoidance". The Definition of the variables is extensive. The methods of assessments and determinants should be left in the main text, the extensive definitions can go to supplementary material preferably as a table.

For example Page 4: "Individuals who withheld from seeking medical care were considered healthcare avoiders.14 Participants were asked whether they had experienced symptoms for which they otherwise would have contacted their GP or medical specialist, but now did not do so because of COVID-19. They were provided with a prespecified list of both symptoms that might have warranted urgent medical assessment and generic symptoms, which made it possible for participants to indicate for which symptoms they had avoided healthcare: palpitations, chest pain, limb weakness, self-perceived cancer-related symptoms (e.g. weight loss, suspicious skin spots), difficulty speaking or facial drooping, vision loss, elevated blood pressure, sudden onset dizziness, dysregulation of diabetes, nausea, fluid retention (oedema), memory complaints, attempts to stop or reduce smoking, and lower back pain. Since lower back pain is generally self-limiting, we specifically used this symptom to contrast with other symptoms that potentially require urgent medical evaluation." This can be included in supplementary material. 

Page 5: "This system enables country-specific divisions of educational levels to be transformed into seven internationally comparable categories.21 Respondents were asked to report their highest level of education attained in the Dutch educational system, after which these results were translated into one of the UNSECO categories, which were eventually merged into four groups. The question 'How do you, in general, appreciate your own health?' was used to assess self-appreciated health, to which respondents could answer 'excellent', 'very good', 'good', 'fair', or 'poor'. We have measured occupational status with the question 'What do you do in everyday life?' with corresponding response categories 'I work (fulltime, part-time, self-employed)', 'I'm on sick leave', 'unemployed', 'retired', or 'other'. Alcohol consumption and smoking status were based on self-reported use during the last 14 days before filling out the questionnaire. Worry about contracting COVID-19 was assessed through the statement 'I worry about contracting COVID-19' with response categories measured on a five-point Likert scale, ranging from 'never' to 'almost continuously'. Respondents were screened for depressive symptoms using ten out of twenty questions from the Center for Epidemiological Studies Depression (CESD) scale, with a weighted maximum score of 29. 22 The higher the score on this scale, the more depressive symptoms participants experienced during the week before completing the questionnaire. Anxiety was measured by seven of out fourteen questions from the Hospital Anxiety and Depression Scale (HADS), which has a weighted maximum score of 20. � This can be included in supplementary material. 

6. Page 7: The prevalence of avoidance increased with age, with 16.9% below age 65 and 29.2% the age of 80 onwards. But per table age 65-79 seems to be the largest avoiders � this needs to corrected. 

7. Page 8, Sensitivity analysis: Please add area under ROC for all the LR models. 

8. Interestingly, poor self-appreciated health had more impact on health care avoidance than worry about contracting COVID-19, OR vs 2.13 vs 1.33 in model-1. This is particular in contrast to what one would assume during the beginning of pandemic. � After the model-1 derivation is explained, it is better to highlight this in discussion portion with re-iteration of Odds ratio. 

9. Do authors want to declare the "bias" related to study in Weakness?

10. May be better to replace 'worry of COVID-19' with 'concern of COVID-19', where appropriate. 

11. Can add line numbers in the text for furthers revisions?

Reviewer #4: What was the time period the participants were answering about their healthcare utilisation? It's not overly clear in methods. Is it 27.02.20-08.04.20 for all patients? Were all participants specifically asked to recall their healthcare utilisation for the same specific time period e.g. 27.02.20-08.04.20? Or was it the month preceding when they received/completed the questionnaire? This time period could be different for some participants depending on when they completed for questionnaire (you say you received the last questionnaire in July for example). It's not overly clear from the text and more detail is probably warranted. Having a standardised time period for all participants seems like it would be best to reduce any bias. If this isn't possible, accounting for what stage the wave was at may be important as toward the start of the wave 1 people may be more likely to avoid, whereas toward the end of wave 1 people may be less likely to avoid due to lower circulating infection or more comfortable with living with the disease and the measures in place. 

Ideally, reporting patients' healthcare utilisation in the same corresponding calendar time period in the previous year/s as when the questionnaire was completed would be good (either in the year before the pandemic e.g. 2019 or many years and average them out e.g. 2014-2019). This may control for seasonal changes in healthcare utilisation as you are only looking at a small time period within 12 months.

Historical information on healthcare utilisation from medical records would be good to see whether patients' healthcare utilisation is consistent over time before the pandemic, rather than basing it on one three month period. Even if this is just descriptively. Showing this dip in healthcare utilisation in a form of graph may be more visually striking than only presenting it in tables/text.

More detail on the methods used for the linkage of GP records would be welcomed for clarity.

Including ethnicity and deprivation status (area [postcode derived] or household deprivation) would be useful here as previous evidence shows that these groups (more deprived and ethnic minorities) are less likely to utilise healthcare/have less access to healthcare. Understanding whether patterns and correlates are the same when including these as covariates would be useful. Further understanding whether there are interactions between healthcare utilisation/avoidance and ethnicity/deprivation would be important as it could be informative. Understanding whether the correlates are similar across these groups would be an important public health message. These groups have generally been found to be at higher risk of COVID-19 mortality too. Not sure if the authors agree, but I think a priori there is enough evidence to examine an interaction between these outcomes. Education and occupation status are proxies for deprivation, but not always the best markers. 

Again, not sure if authors will agree, but an analysis examining the determinants of healthcare avoidance among participants with specific chronic diseases (rather than all grouped together as in the supplementary data) may be interesting here. Seeing whether there were specific disease states where people avoided healthcare more/if the correlates of these were similar between disease states would be important as not all disease states are the same. 

Follow-up associations with outcomes? I think this would addition would strengthen this paper and also contextualise what the healthcare avoidance actually has meant in real terms. The work undertaken so far is still important though.

Multivariable and not multivariate in the 'Determinants related to healthcare avoidance' section. I'm guessing the outcome was only measured once and not repeated measures.

Also, if looking at the determinants/correlates you will be taking the coefficient of each of the covariates in your model, rather than the coefficient of exposure of interest. In that case, checking for collinearity is important here so will need to clarify that in text and accordingly deal with any collinear variables in model.

In limitations need to add in text around the generalisability of the results as these are only in older individuals and any other biases in the cohort from the original study (e.g. could be more affluent than average, more educated, healthier etc).

[LINK]

---

## [Decision Letter · Decision Letter 2]

18 Oct 2021

Dear Dr. Splinter,

Thank you very much for re-submitting your manuscript "Prevalence and determinants of healthcare avoidance during the COVID-19 pandemic: a population-based cross-sectional study" (PMEDICINE-D-21-02940R2) for review by PLOS Medicine.

I have discussed the paper with my colleagues and the academic editor and it was also seen again by three reviewers. I am pleased to say that provided the remaining editorial and production issues are dealt with we are hoping to accept the paper for publication in the journal.

[LINK]

We look forward to receiving the revised manuscript by Oct 25 2021 11:59PM.   

Sincerely,

Callam Davidson, 

Associate Editor 

PLOS Medicine

plosmedicine.org

Requests from Editors:

Line 70: Please update to ‘which determinants are associated with this behaviour’

Line 80: Please replace ‘related to’ with ‘associated with’

Line 85-86: Please update to ‘Importantly, our findings suggest this behaviour may be associated with certain vulnerable groups within the population’

Lines 87-90: Please consider replacing this bullet (which focuses on future studies) with one that briefly discusses study limitations.

Lines 91-93: The findings of this study can be used to develop policy interventions targeted to vulnerable individuals who may be more likely to exhibit health avoidance behaviours.

Line 108: Please replace ‘timely presenting themselves to’ with ‘seeing’

Line 122: Please delete ‘because it allows to characterise the unseen population’ providing you do not feel it alters meaning (to me, it appears redundant).

Line 128: Causality should not be overstated given the cross-sectional design – please update to ‘sought to determine which potential determinants were associated with healthcare avoidance.

Line 142: Please update to ‘This study is reported as per the Strengthening the Reporting of Observational Studies in Epidemiology (STROBE) guideline (S1 Checklist).’

Line 143-144 and line 168-169: These lines stating ethical approval/consent can be deleted here as you repeat this information again below.

Line 195: Update ‘inquired’ to ‘asked’.

Lines 245 and 247: Update to ‘were of a lower educational level’

Line 249-250: Update to ‘Most of those participants considered to be healthcare avoiders’.

Lines 290-1 and line 321: Update ‘stronger related’ to ‘more strongly associated with’.

Line 298: Update to ‘In this cross-sectional study’

Lines 301-303: Please update to ‘In multivariable analyses, we have shown that female sex, low self-appreciated health, and high levels of anxiety and depression were associated with healthcare avoidance during the COVID-19 pandemic.’

Line 344-346: Please update to ‘findings of our study suggest that healthcare avoidance during COVID-19 may be prevalent amongst those who are in greater need of it in the population, such as older individuals, those with low perceived health and those who report symptoms of poor mental health.’

Line 358: Please update to ‘Future studies should examine healthcare seeking behaviour among ethnically diverse populations which remain understudied.’

Prospective Analysis Plan: Thank you for providing your prospective analysis plan. Please ensure any changes in the analysis-- including those made in response to peer review comments—have been identified as such in the Methods section of the paper, with rationale.

Comments from Reviewers:

Reviewer #2: The authors have addressed my concerns and I now recommend publication.

Peter Flom

Reviewer #3: Thank you for addressing the comments and editing the manuscript. 

[LINK]

---

## [Editor Report · Decision Letter 3]

26 Oct 2021

Dear Dr Licher, 

On behalf of my colleagues and the Academic Editor, Dr Sanjay Basu, I am pleased to inform you that we have agreed to publish your manuscript "Prevalence and determinants of healthcare avoidance during the COVID-19 pandemic: a population-based cross-sectional study" (PMEDICINE-D-21-02940R3) in PLOS Medicine.

When making these updates, please also make the following changes:

* Lines 89-90: Please delete the sentence 'Moreover, we did not systematically inquire self-perceived severity of reported symptoms'

* Lines 223-225: Please update this sentence to 'Finally, as a result of the peer review process, we have additionally stratified the analyses between the self-reported chronic diseases included in this study to examine whether the strength of the associations would differ depending on the type of disease'

PRESS

Sincerely, 

Callam Davidson 

Associate Editor 

PLOS Medicine